# The Role of Fibrinolytic Regulators in Vascular Dysfunction of Systemic Sclerosis

**DOI:** 10.3390/ijms20030619

**Published:** 2019-01-31

**Authors:** Yosuke Kanno

**Affiliations:** Department of Clinical Pathological Biochemistry, Faculty of Pharmaceutical Science, Doshisha Women’s College of Liberal Arts, 97-1 Kodo Kyo-tanabe, Kyoto 610-0395, Japan; ykanno@dwc.doshisha.ac.jp; Tel.: +81-0774-65-8629

**Keywords:** Fibrinolytic regulators, SSc, vascular dysfunction

## Abstract

Systemic sclerosis (SSc) is a connective tissue disease of autoimmune origin characterized by vascular dysfunction and extensive fibrosis of the skin and visceral organs. Vascular dysfunction is caused by endothelial cell (EC) apoptosis, defective angiogenesis, defective vasculogenesis, endothelial-to-mesenchymal transition (EndoMT), and coagulation abnormalities, and exacerbates the disease. Fibrinolytic regulators, such as plasminogen (Plg), plasmin, α2-antiplasmin (α2AP), tissue-type plasminogen activator (tPA), urokinase-type plasminogen activator (uPA) and its receptor (uPAR), plasminogen activator inhibitor 1 (PAI-1), and angiostatin, are considered to play an important role in the maintenance of endothelial homeostasis, and are associated with the endothelial dysfunction of SSc. This review considers the roles of fibrinolytic factors in vascular dysfunction of SSc.

## 1. Introduction

Systemic sclerosis (SSc) is an autoimmune rheumatic disease of unknown etiology that is characterized by vascular dysfunction and fibrosis of the skin and visceral organs as well as peripheral circulatory disturbance [1]. This process usually occurs over many months and years and can lead to organ dysfunction or death. 

In SSc, vascular disorders are observed from early onset to the appearance of late complications and affect various organs, including the lungs, kidneys, heart, and digital arteries, and exacerbate the disease [2]. Microvascular disorders, such as Raynaud’s phenomenon, telangiectasias, and digital ulcers, frequently occur in SSc patients [2,3,4]. In contrast, macrovascular disorders, such as those of the coronary arteries, are rarely involved in SSc [2,5,6]. In SSc, the vascular dysfunction is caused by vascular and endothelial cell (EC) injury, defective angiogenesis, defective vasculogenesis, endothelial-to-mesenchymal transition (EndoMT), vascular tone alteration, and coagulation abnormalities [7], and is associated with abnormalities in the immune system, such as T-cells, B-cells, mast cells, macrophages infiltration, immune activation, and auto-antibody production, as well as abnormalities in the extracellular matrix (ECM) metabolism, such as myofibroblast differentiation, ECM over-production, and the inhibition of ECM degradation. These abnormalities may influence each other and lead to the development of pulmonary arterial hypertension (PAH) and fibrosis [2] (Figure 1). However, the detailed mechanism underlying the relationship between “fibrosis” and “vascular dysfunction” remains unclear. It is reported that vasculopathy occurs in various mice, as urokinase-type plasminogen activator receptor (uPAR)-deficient mice develop EC apoptosis and severe loss of micro-vessels [8]. Caveolin-1-deficient mice show dilated cardiomyopathy and pulmonary hypertension [9]. Caveolin-1 is associated with the internalization and degradation of transforming growth factor-β (TGF-β) receptors and regulates TGF-β signaling [10]. Fli1-deficient mice show a disorganized dermal vascular network with greatly compromised vessel integrity and increased vessel permeability and impaired vascular homeostasis. Fli1 is associated with the expression of platelet/endothelial cell adhesion molecule (PECAM)-1, platelet derived growth factor (PDGF), and sphingosine-1-phosphate receptors (S1PR) [11]. Fos-related antigen-2 (Fra-2) transgenic mice develop microvascular and proliferative vasculopathy, and pulmonary vascular lesions resembling SSc-associated PAH [12]. However, while these factors may play a critical role in the onset of SSc-associated vascular disorders, the detailed mechanism underlying their involvement is unclear.

The fibrinolytic system dissolves fibrin and maintains vascular homeostasis. The regulators of fibrinolysis contain plasminogen (Plg) a proenzyme, which is converted to the active serine protease plasmin, a main component of the fibrinolytic system, through the action of a tissue-type plasminogen activator (tPA) or urokinase-type plasminogen activator (uPA) and uPA receptor (uPAR). In contrast, alpha2-antiplasmin (α2AP) functions as the main inhibitor of plasmin, resulting in the formation of the stable inactive complex plasmin-α2AP and the inhibition of fibrinolysis [13]. Plasminogen activator inhibitor-1 (PAI-1) binds and blocks tPA and uPA and inhibits the conversion of Plg to plasmin [14]. In addition, angiostatin is a circulating inhibitor of angiogenesis generated by the proteolytic cleavage of Plg. These fibrinolytic regulators have various functions, such as growth factor and matrix metalloproteinase (MMP) activation, ECM degradation, and fibrinolysis (Figure 2). It is reported that ECs synthesize tPA, uPA, uPAR, and PAI-1, and that fibrinolytic regulators play an important role in the maintenance of endothelial homeostasis [15,16,17,18,19,20]. The levels of plasmin-α2AP complex and D-dimer in plasma are elevated in SSc [21,22,23] and the expression of α2AP is elevated in fibrotic tissue of SSc model mice and dermal fibroblasts obtained from patients with SSc [24,25]. α2AP deficiency attenuates the development of fibrosis in SSc model mice [26,27] and uPAR deficiency promotes the development of fibrosis [28]. In addition, the levels of uPA, soluble uPAR (suPAR), tPA, PAI-1, and angiostatin are elevated in SSc [29,30,31,32]. Furthermore, uPAR-deficient mice develop vasculopathy [8]. α2AP induces vascular injury, and α2AP deficiency attenuates the SSc-associated vascular dysfunction in SSc model mice [33]. These fibrinolytic regulators may be associated with the SSc-associated vascular disorders. 

This review focuses on the role of fibrinolytic regulators in the vascular dysfunction of SSc.

## 2. The Various Functions of Fibrinolytic Regulators

### 2.1. Plasminogen (Plg) and Plasmin

Plg is converted to the active serine protease plasmin, a main component of the fibrinolytic system, by tPA or uPA/uPAR [34]. Plg is a single-chain glycoprotein that consists of an N-terminal activation peptide and five kringle domains and is synthesized by liver cells [34]. Plg can bind not only fibrin, but also to various receptors, such as the heterotetrametric complex Annexin A2-S100A10, enolase-1, histone H2B, and the plasminogen receptor Plg-RKT [35]. On binding to Annexin A2-S100A10, Plg is associated with the progression of inflammation, thrombosis, cancer, and autoimmune diseases [35,36,37]. Histone H2B contributes to the Plg-binding capacity of cells and tethers to the surface of cells by interacting with phosphatidylserine on differentiated or apoptotic monocytoid cells [38,39]. Enolase-1 can bind to Plg at the cell surface and promote plasmin production and monocyte migration [40]. Plg-RKT is involved in the Plg-dependent regulation of macrophage invasion, chemotactic migration, and recruitment in the inflammatory response [41]. Plg/plasmin regulates the activation of growth factors, such as TGF-β, basic fibroblast growth factor (bFGF), vascular endothelial growth factor (VEGF), insulin-like growth factor-binding protein 5 (IGFBP-5), and pro-brain derived neurotrophic factor (proBDNF), as well as the activation of MMPs, such as MMP-1, MMP-3, and MMP-9, and ECM (collagen, fibronectin, laminin, entactin, tenascin, thrombospondin, and perlecan) degradation [15,34,42,43,44]. Plasmin also activates protease-activated receptor (PAR)-1 and PAR-4 factors V, VIII, and X, and induces gene expression, pro-coagulant effects, and platelet activation [44,45]. Furthermore, Plg can bind to the central complement protein C3, the C3 cleavage products C3b, C3d, and C5, as well as affect complement action [46]. The areas of involvement of plasmin include cell migration, cell proliferation, monocyte chemotaxis, neutrophil aggregation, and the inflammatory response through various signal pathways, as well as tissue remodeling, wound healing, angiogenesis, cancer, bone metabolism, and glucose metabolism [42,43,47,48]. 

### 2.2. α2-Antiplasmin (α2AP)

α2AP is a serine protease inhibitor (serpin) with a molecular weight of 65 to 70 kDa [13] that rapidly inactivates plasmin in fibrin clots or in the circulation, resulting in the formation of a stable inactive complex, plasmin-α2AP [49]. The N-terminal sequence is crosslinked to fibrin by factor XIIIa, whereas the C-terminal region mediates the initial interaction with plasmin. A protease, such as antiplasmin-cleaving enzyme (APCE) or fibroblast activation protein (FAP), causes the conversion of Met-α2AP to Asn-α2AP (12-amino-acid residue shorter form) [50,51]. α2AP mRNA is detected in a number of murine tissues, such as the liver, kidney, intestine, spleen, lung, muscle, ovary, testis, cerebral cortex, hippocampus, cerebellum, bone, skin, and placenta [52]. α2AP is known to regulate angiogenesis, inflammation responses, cell proliferation, differentiation, the recruitment of lymphocytes and neutrophils, wound healing, vascular remodeling, fibrosis, bone formation, and brain functions, and also acts as a plasmin inhibitor [25,53,54,55,56,57,58]. The α2AP N-terminal region is composed of three β-sheets and nine α-helices [59]. α2AP is most closely related to the non-inhibitory serpin pigment epithelium-derived factor (PEDF), showing a markedly similar structure [60,61]. α2AP can bind and activate the PEDF receptor adipose triglyceride lipase (ATGL)/calcium-independent phospholipase A2 (iPLA2) and induce cytokine production, ECM production, cell differentiation, and cell proliferation [27,62]. α2AP also contains an RGD sequence, which is a sequence for cell recognition through integrins, and thereby may regulate integrin signaling [63]. 

### 2.3. Urokinase-Type Plasminogen Activator (uPA) and Its Receptor (uPAR)

uPA is a serine protease that causes the conversion of Plg to plasmin. The N-terminal domain of uPA, known as the N-terminal fragment (ATF), can bind to its receptor, uPAR. In contrast, the C-terminal domain of uPA is associated with catalytic activity [20]. uPAR is a glycosylphosphatidylinositol (GPI)-anchored protein composed of three domains (D1, D2, and D3). [64]. uPAR can interact with a number of proteins, including uPA, integrins, vitronectin (Vn), and low-density lipoprotein receptor-related protein (LRP-1), in the membrane and regulate various signaling pathways [64,65]. uPAR is cleaved between the D1 and D2 domains and the GPI-anchor domain by various enzymes, including uPA, plasmin, MMP-3, MMP-12, MMP-19, MMP-25, GPI-specific phospholipase D, and cathepsin G, to form soluble uPAR (suPAR; full length D1-D3, D2D3, and D1) [66]. suPAR activates the G protein-coupled receptor N-formyl-Met-Leu-Phe (FPRL1) and regulates vascular smooth muscle cell (VSMC) migration, the recruitment of monocytes, stem cell mobilization, and leukocyte trafficking [66,67,68]. suPAR is also associated with thrombosis and the inhibition of plasmin generation [69]. uPA and uPAR are involved in not only cell surface plasmin generation, but also in the promotion of various intracellular signaling pathways via interaction with transmembrane proteins, such as integrins and the mediation of cellular adhesion, differentiation, proliferation, and migration [20,70,71,72]. uPA and uPAR regulate cell growth, inflammatory reaction, immune response, tissue remodeling, angiogenesis, adipose tissue development, fibrosis, bone metabolism, and glucose metabolism, and are associated with the pathogenesis of various diseases, such as rheumatoid arthritis, periodontitis, diabetes, cancer, and fibrosis [17,20,28,70,71,72,73,74]. 

### 2.4. Tissue-Type Plasminogen Activator (tPA)

tPA is secreted from ECs and can convert Plg into plasmin. tPA is a mosaic protein composed of five distinct modules: A finger domain, an epidermal growth factor (EGF)-like domain, two kringle domains, and a serine protease proteolytic domain [75]. The finger domain can bind to fibrin, the EGF-like domain is associated with the hepatic recapture of tPA, and the kringle domains are associated with the binding and activation of substrates and/or receptors, such as Plg, PDGF, and N-methyl-d-aspartate receptor (NMDAR) [75]. tPA also regulates MMP activation, LRP-1 or NMDAR interaction, ECM remodeling, and growth factor activation, such as BDNF, angiogenesis, neurogenesis, and adenylate cyclase activation [76].

### 2.5. Plasminogen Activator Inhibitor-1 (PAI-1)

PAI-1 is a serpin that inhibits tPA and uPA and regulates the plasmin activation and the fibrinolytic system [77]. PAI-1 is synthesized in a number of cells, including ECs, adipocytes, macrophages, cardiomyocytes, fibroblasts megakaryocytes, hepatocytes, and platelets [78]. PAI-1 is composed of three β-sheets and nine α-helices and can bind to the somatomedin B domain of Vn, interact with the α-3 subunit of proteasome, and interfere with cell adhesion to the ECM [78,79]. The expression of PAI-1 is induced by various factors, including TGF-β, bFGF, interleukin-1β (IL-1β), tumor necrosis factor-α (TNF-α), EGF, insulin-like growth factor 1 (IGF-1), and PDGF [79,80,81,82,83]. PAI-1 is associated with the development of a number of diseases, such as thrombosis, atherosclerosis, endometriosis, cancer, obesity, insulin resistance, diabetes, fibrosis, and cardiovascular disease [78].

### 2.6. Angiostatin

Angiostatin is an internal fragment of Plg generated by the proteolytic cleavage of Plg [84]. Angiostatin includes the four kringle domains of Plg, which perform an anti-angiogenesis function. The generation of angiostatin is associated with uPA, tPA, elastase, and MMP [85,86,87,88]. Angiostatin inhibits EC proliferation, EC migration, and tube formation, induces EC apoptosis, and attenuates VEGF expression by binding to ATP synthase, angiomotin, integrins, and annexin II or by preventing G2/M transition [89,90,91,92]. In addition, angiostatin induces the production of other anti-angiogenic factors, such as thrombospondin-1 [92]. Angiostatin also inhibits neutrophil activation and migration [93], monocyte and macrophage migration [94], and leukocyte recruitment and has an anti-inflammatory function [95]. It inhibits tumor cell invasion by blocking plasminogen binding to CD26 [96] and inhibits MMP expression in ECs [97].

## 3. The Role of Fibrinolytic Regulators in Vascular and EC Injury in SSc

Vascular and EC injury is an early and initiating event in SSc. A number of factors (e.g., infections, cytotoxic T-cells, oxidative stress, auto-antibodies, ischemia-reperfusion) cause persistent EC activation and stimulate the production of various cytokines, EC apoptosis, impairment of cell-cell adhesion, and the activation of complement and coagulant pathways [98]. In addition, these factors also induce the production of vasodilators, such as nitric oxide (NO), vasoconstrictors, such as endothelin-1 (ET-1), and platelet activation, and lead to the impairment of vascular tone control and vascular and EC damage [2,98,99,100,101]. 

It is reported that Plg induces EC apoptosis [102]. Plasmin also damages the endothelial barrier function and EC integrity and induces EC injury [103]. Plasmin is known to regulate the vascular endothelial function and influence the progression of various cardiovascular diseases through fibrinolysis, the degradation of the ECM, and MMP and TGF-β activation [104,105]. Furthermore, plasmin regulates the fibrin-mediated EC spread and proliferation [106], MMP-mediated cell adhesion and cell migration [107], and TGF-β-induced EC apoptosis [108]. These direct and indirect effects of plasmin may be associated with the maintenance of the endothelial function. Conversely, uPA inhibits EC apoptosis through the induction of X-linked inhibitor of apoptosis protein [109]. uPAR is involved in the high-molecular-weight kininogen (HKa)-mediated apoptoic effect [110]. α2AP induces vascular damage, such as the reduction of blood vessels and blood flow in mice, and α2AP neutralization improves vascular damage in SSc model mice [33]. In addition, α2AP is associated with vascular remodeling and EC apoptosis [57]. PAI-1 reportedly induces EC apoptosis, but protects against FasL-mediated apoptosis [111,112]. Angiostatin regulates the inhibition of EC proliferation, EC migration, and tube formation, as well as the induction of EC apoptosis [89,90,91,113]. In SSc, the changes in the expression of the fibrinolytic regulators may regulate the endothelial function and dysfunction.

## 4. The Role of Fibrinolytic Regulators in Defective Angiogenesis in SSc

In SSc, angiogenesis is incomplete or lacking despite the increased expression of the pro-angiogenic factor VEGF [114]. VEGF plays a critical role in the maintenance of vascular functions, such as EC growth, activation, proliferation, and migration, through the VEGFR2 signal transduction pathways and also regulates angiogenesis [115]. The expression of VEGF is elevated in various cells, such as fibroblasts, ECs, and immune cells, but vascular insufficiency manifests in SSc [116,117]. The impairment of VEGF responses may cause vascular dysfunction in SSc, but the detailed mechanisms remain unclear. 

Plasmin is known to regulate vascular endothelial functions and influence the progression of various cardiovascular diseases through fibrinolysis, the degradation of matrix proteins, and the activation of growth factors [104]. In addition, VEGF can be processed by plasmin and thereby released from the ECM [118,119]. α2AP attenuates the VEGF-induced pro-angiogenic effects, such as tube formation and EC proliferation, by blocking the VEGFR2 signal pathway in ECs [33]. In addition, α2AP is associated with VEGF production in fibroblasts and angiogenesis [53]. In SSc, fibroblasts are likely to be important effector cells. SSc fibroblasts inhibit angiogenesis and induce vascular dysfunction [1,33,120]. The blocking of α2AP markedly improves the SSc dermal fibroblast-induced vascular dysfunction, indicating that SSc fibroblast-derived α2AP affects vascular dysfunction in the disease [33]. An increased α2AP expression in SSc may cause impairment of the VEGF response and lead to vascular dysfunction. uPA and uPAR play important roles in angiogenesis and modulate the VEGF signaling [121,122]. uPA and uPAR are associated with the impairment of angiogenesis in SSc, and the SSc EC-conditioned medium attenuates uPA-dependent EC proliferation and invasion. In addition, the cleavage of uPAR by the overproduction of MMP-12 in SSc inhibits angiogenesis [120,123]. uPAR can interact with integrins, which mediate actin assembly in ECs and are associated with angiogenesis and vascular alterations in SSc [124,125,126]. uPAR also regulates VSMC proliferation and migration [127,128]. PAI-1 inhibits the binding of VEGFR-2 to β3 integrin as well as VEGF signaling [129]. In addition, PAI-1 binds to uPA and uPAR to exert anti-angiogenic effects [130]. tPA induces VEGF production through the ERK and p38 pathways in ECs [131].

Angiopoietins regulate vascular homeostasis through the Tie2 receptor [132,133,134]. Angiopoietin-1 (Ang-1) mediates vascular remodeling and stabilization, while angiopoietin-2 (Ang-2) functions as a Tie2 agonist or antagonist and is associated with angiogenesis and vascular permeability [133,135]. Ang-1 is decreased while Ang-2 is increased in the sera of patients with SSc and the differential expression of Ang-1/Ang-2 may be associated with the progression of SSc [136]. tPA regulates Ang-2 production [137], so an increase in tPA may induce an increase in Ang-2. In addition, α2AP inhibits the Ang-1-induced EC sprouting [138], and the suppression of uPA and uPAR inhibits Tie2 activation and attenuates angiogenesis [139]. Ang-1 or Tie2 can interact with integrins [140,141]. α2AP or uPA/uPAR-mediated Tie2 activation may be associated with the binding of integrins.

Angiostatin is known to be an anti-angiogenic factor that regulates EC proliferation, EC migration, EC apoptosis, and VEGF expression while inhibiting angiogenesis [89,90,91,92]. Angiostatin is generated by elastase [84]. MMP-12 is a macrophage elastase, and MMP-12 is elevated in SSc [120]. This increase in the MMP-12 expression may cause angiostatin overproduction, thereby leading to defective angiogenesis.

## 5. The Role of Fibrinolytic Regulators on EPC Functions

Vasculogenesis is the generation of new blood vessels through the differentiation of pericytes and the recruitment and differentiation of bone marrow-derived endothelial progenitor cells (EPCs) [98]. After vascular damage, EPCs are mobilized from the bone marrow to differentiate into ECs or VSMCs [2]. Although the role of EPCs in SSc vasculopathy is unclear, they are reportedly detected in the peripheral blood of SSc patients [142,143]. Fibrinolytic regulators are associated with EPC-mediated sprouting angiogenesis [144]. tPA enhances the mobilization of EPCs from bone marrow [145,146]. Increased uPA expression regulates EPC migration [147]. The recruitment of uPAR in caveolar-lipid rafts regulates EPC-mediated neovascularization [148,149], and angiostatin inhibits EPC-mediated neovascularization [150].

## 6. The Role of Fibrinolytic Regulators in EndoMT in SSc

Recent studies suggest that EndoMT is a type of transdifferentiation by which ECs lose their specific morphology/markers and acquire myofibroblast-like features. EndoMT is associated with the progression of vascular dysfunction in SSc [7,151,152]. EndoMT plays an important role in the development of SSc-associated interstitial lung disease (ILD), PAH, and fibrosis [153]. It is reported that EndoMT is induced by inflammatory responses and results in the fibrotic changes [154]. EndoMT exhibits features similar to those of epithelial-to-mesenchymal transition (EMT) and is induced by cytokines and growth factors, such as TGF-β, IL-1 β, TNF-α, ET-1, Notch, and Wnt, as well as hypoxia [154,155]. The conversion of ECs by EndoMT may cause not only vascular dysfunction, but also the development of fibrosis, which exacerbates the disease severity. α2AP induces the production of TGF-β, IL-1β, and TNF-α [25,27,62,156], as well as myofibroblast differentiation through EMT [24,25,62]. α2AP may be associated with the onset of EndoMT in SSc, and uPAR deficiency also promotes EndoMT [152]. In addition, uPAR is associated with EMT [157,158] and myofibroblast differentiation [159]. uPA/uPAR regulate inflammatory responses through various signal pathways [48,160,161]. Similarly, caveolin-1 deficiency also induces EndoMT and is associated with the development of fibrosis [162]. Caveolin-1 regulates the uPA expression and uPAR-mediated signaling [163,164], and the uPA/uPAR-mediated cell signaling may regulate the progression of EndoMT. ET-1 and Wnt reportedly regulate PAI-1 production [165,166], and PAI-1 deficiency is shown to promote EndoMT [167]. Furthermore, fibrinolytic regulator-mediated growth factor activation and MMP activation may be associated with EndoMT and play important roles in the EndoMT-mediated progression of SSc.

## 7. The Role of Fibrinolytic Regulators in Coagulation Abnormalities in SSc

Microvascular thrombosis and fibrin deposition were observed in patients with SSc, and an imbalance in coagulation and fibrinolysis causes vascular damage [2,99,168]. The levels of von Willebrand factor (vWF), fibrinogen, ET-1, sphingosine-1-phosphate (S1P), and lysophosphatidic acid (LPA) are elevated in SSc [2,99]. In addition, a specific nonintegrin receptor for type I collagen was found to be elevated in platelets obtained from SSc patients, and an increased responsiveness of SSc platelets to 5-hydroxytryptamine (5HT), adrenaline, ADP, and collagen were reported [169,170]. Those increases may cause the activation of platelets and hypercoagulation. Furthermore, plasmin induces platelet activation, platelet aggregation, and platelet release reaction through PAR [171,172,173]. Plasmin also enhances their sensitivity to ADP [173]. In SSc, increases in the levels of uPA and tPA may promote plasmin generation and the activation of platelets, which synthesize and release α2AP and PAI-1 [174,175].

The expression of α2AP and PAI-1 [24,31] and uPAR cleavage by MMP-12 overexpression [120] is elevated in SSc. Furthermore, α2AP can be crosslinked to the fibrin surface by activated FXIIIa [63], and PAI-1 binds to fibrin through Vn [176]. The inactivation of plasmin by increases in the expression of α2AP and PAI-1 may cause the impairment of fibrinolysis. In addition, Barrett et al. suggest that the angiostatin generation induced by elastase-degraded Plg may underlie the fibrinolytic shutdown [87]. These changes in fibrinolytic regulators may cause the impairment of fibrinolysis and lead to the deposition of fibrin and coagulation abnormalities characteristic of SSc.

## 8. The Role of Fibrinolytic Regulators in Vascular Tone Alteration in SSc

In SSc, it has been reported that the eNOS expression and NO release are decreased, and the impairment of NO response attenuates vasodilation [99]. Conversely, vasoconstrictors, such as ET-1, are elevated in SSc and cause abnormal vasoconstriction [99]. These changes in the vascular tone in SSc may lead to vascular damage. tPA, PAI-1, and plasmin inhibitor have been reported to modulate vasodilation and vasoconstriction and regulate the vascular tone [177,178]. In addition, PAI-1 deficiency prevents hypertension in response to long-term NOS inhibition [179], and uPA promotes the LRP-mediated eNOS activation [180]. Furthermore, angiostatin inhibits the VEGF-induced NO production and is involved in vasodilation [181,182]. The fibrinolytic system may be involved in the vascular tone alterations observed in SSc.

## 9. The Effect of Fibrinolytic Regulators on SSc-Associated PAH

SSc-associated PAH is a leading cause of death in SSc, with a prevalence of around 10% and a three-year mortality rate of 50% [183,184]. Although the mechanisms underlying the onset of SSc-associated PAH remain unclear, it is believed that inflammation and vascular injury-mediated pulmonary vascular remodeling are involved [183]. uPAR is reportedly involved in SSc-associated PAH [185]. tPA is elevated, PAI-1 is decreased, and the ratio of uPA and PAI-1 is decreased in the bronchoalveolar lavage fluid (BALF) in idiopathic pulmonary fibrosis patients with pulmonary hypertension (PH) [186]. In addition, Plg and uPA deficiency protect against the development of hypoxia-induced PAH, and uPA-generated plasmin is associated with the onset of PH [187]. Furthermore, the levels of platelet angiostatin are elevated in PAH patients [188]. Angiostatin also aggravates PH in chronically hypoxic mice [189]. These fibrinolytic regulators may play an important role in the onset of SSc-associated PAH.

## 10. Conclusion and Therapeutic Perspectives

In SSc, vascular dysfunction is linked to the innate and adaptive immune systems and fibrosis and plays an important role in the development of immune abnormalities, auto-antibody production, ECM deposition, and fibrosis. The increase, inhibition, and degradation of these fibrinolytic regulators in SSc may cause vascular and EC injury, defective angiogenesis, defective vasculogenesis, EndoMT, impaired fibrinolysis, coagulation abnormalities, vascular tone alteration, and SSc-associated PAH. The fibrinolytic regulators directly or indirectly mediate the endothelial functions through fibrinolysis, cell migration, differentiation, proliferation, cytokines production, growth factor activation, MMP activation, ECM degradation, and the regulation of various signal pathways, and may be associated with the vascular alteration and dysfunction observed in SSc (Figure 3). The various functions of fibrinolytic regulators play a critical role in the pathogenesis of SSc, making these factors potential therapeutic targets for SSc. The regulation of fibrinolytic regulator-initiated pathways may be a novel therapeutic approach to SSc.

## Figures and Tables

**Figure 1 ijms-20-00619-f001:**
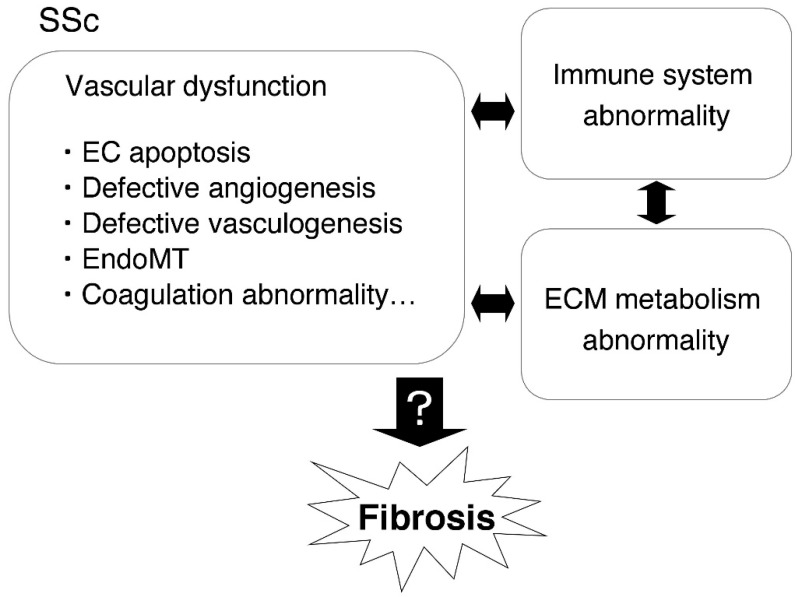
Vascular dysfunction in systemic sclerosis (SSc). In SSc, the vascular dysfunction is caused by vascular and endothelial cell (EC) injury, defective angiogenesis, endothelial-to-mesenchymal transition (EndoMT), and coagulation abnormalities, and is associated with abnormalities in the immune system and extracellular matrix (ECM) metabolism. These abnormalities may induce myofibroblast differentiation, ECM deposition, and the development of fibrosis.

**Figure 2 ijms-20-00619-f002:**
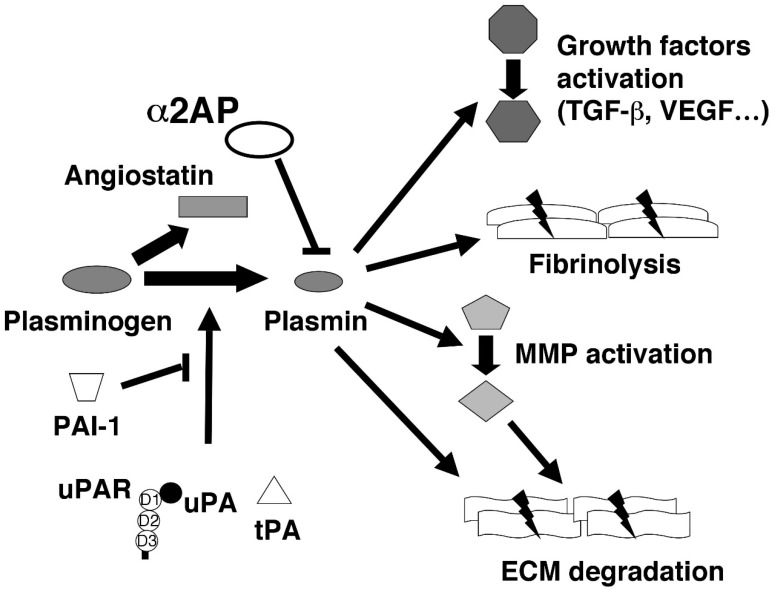
The functions of fibrinolytic regulators. The fibrinolytic system contains plasminogen (Plg), which is converted to plasmin, a main component of the fibrinolytic system, through the action of a tissue-type plasminogen activator (tPA) or urokinase-type plasminogen activator (uPA) and uPA receptor (uPAR). In contrast, α2AP and PAI-1 function as the main inhibitor of Plg/plasmin system. Plg is also converted to angiostatin. These fibrinolytic regulators have various functions, such as fibrinolysis, growth factors, matrix metalloproteinase (MMP) activation, and ECM degradation.

**Figure 3 ijms-20-00619-f003:**
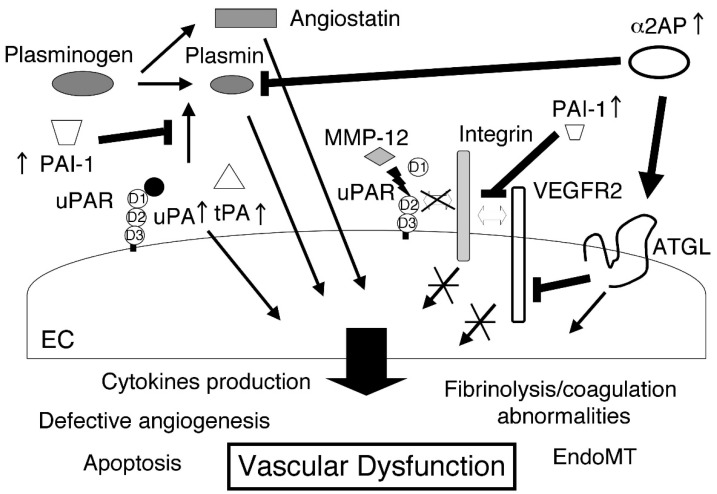
The role of fibrinolytic regulators in SSc. In SSc, the increase, inhibition, and degradation of these fibrinolytic regulators regulate cytokines production and various signal pathways. The various functions of fibrinolytic regulators may cause vascular and EC injury, defective angiogenesis, EndoMT, and coagulation abnormalities, and lead to vascular alteration and dysfunction.

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
