# Peer review of "The Role of Fibrinolytic Regulators in Vascular Dysfunction of Systemic Sclerosis"

_ijms, 2019, doi:10.3390/ijms20030619_

Round 1
Reviewer 1 Report
this is a solid and thorough review
some suggestions
ref 1 should be substituted with a recent paper
Pathogenesis of systemic sclerosis: recent insights of molecular and cellular mechanisms and therapeutic opportunities John Varga, Maria Trojanowska, Masataka Kuwana J scleroderma relat disord 2017; 2(3): 137 - 152
ref 169 Cerinic MM= matucci cerinic mm
please revise carefully the references as some are repeated
Author Response
1. ref 1 should be substituted with a recent paper. Pathogenesis of systemic sclerosis: recent insights of molecular and cellular mechanisms and therapeutic opportunities John Varga, Maria Trojanowska, Masataka Kuwana J scleroderma relat disord 2017; 2(3): 137 – 152.
Thank you for your comment. I searched for “John Varga, Maria Trojanowska, Masataka Kuwana J scleroderma relat disord 2017; 2(3): 137–152.” in Pubmed. However, I could not find this paper in Pubmed. Therefore, I could not check this paper.
2. ref 169 Cerinic MM= matucci cerinic mm.
As pointed out, I corrected it.
3. Please revise carefully the references as some are repeated
Thank you for your comment. I checked this paper again.
In addition, this manuscript was checked by a professional English editing service.
Reviewer 2 Report
The review titled “the role of fibrinolytic regulators in vascular dysfunction of systemic sclerosis” by Yosuke Kanno
1. The figure 1: “ tissue fibrosis” has not cited, and its role in vascular dysfunction could be reviewed
2. The mechanism of tissue fibrosis and role of fibrinolystic factors/thrombosis in vascular dysfunction could be described.
3. There are several typos and grammetical errors in the text could be fixed
Author Response
1. The figure 1: “ tissue fibrosis” has not cited, and its role in vascular dysfunction could be reviewed.
Thank you for your comment. Vascular dysfunction plays an important role in the pathogenesis of SSc, and it has been reported that vascular dysfunction is associated with the development of fibrosis in SSc. In SSc, vascular dysfunction may directly or indirectly induce the development of fibrosis. However, the detailed mechanism of relationship between “fibrosis” and “vascular dysfunction” remains unclear. Therefore, I added “?” in Figure 1, and added the comment “These abnormalities may influence each other and lead to the development of pulmonary arterial hypertension (PAH) and fibrosis [2] (Fig. 1). Furthermore, I added the comment “However, the detailed mechanism underlying the relationship between “fibrosis” and “vascular dysfunction” remains unclear.” (Page 3 line 17 to Page 4 line 3).
2. The mechanism of tissue fibrosis and role of fibrinolystic factors/thrombosis in vascular dysfunction could be described.
Thank you for your comment. Vascular dysfunction plays an important role in the pathogenesis of SSc, and it has been reported that vascular dysfunction is associated with the development of fibrosis in SSc. In SSc, vascular dysfunction may directly or indirectly induce the development of fibrosis. However, the detailed mechanism of relationship between “fibrosis” and “vascular dysfunction” remains unclear. Therefore, I added the comment “However, the detailed mechanism underlying the relationship between “fibrosis” and “vascular dysfunction” remains unclear.” (Page 3 line 17 to Page 4 line 3).
3. There are several typos and grammetical errors in the text could be fixed
Thank you for your comment. This manuscript was checked by a professional English editing service.
Reviewer 3 Report
The review paper is quite interesting.
Minor points that have to be corrected:
Fig 1: please correct abnormality
L82: correct "is consists"L122: correct "integrins"
L124:capitalize "urokinase"
Author Response
1. Fig 1: please correct abnormality.
Thank you for your comment. As pointed out, I corrected it.
2. L82: correct "is consists"L122: correct "integrins"
As pointed out, I corrected them.
3. L124:capitalize "urokinase"
As pointed out, I corrected them.
In addition, this manuscript was checked by a professional English editing service.